# Determinants of Place of Delivery during the COVID-19 Pandemic—Internet Survey in Polish Pregnant Women

**DOI:** 10.3390/medicina58060831

**Published:** 2022-06-20

**Authors:** Mateusz Strózik, Lukasz Szarpak, Ishag Adam, Jacek Smereka

**Affiliations:** 1Department of Emergency Medical Service, Wroclaw Medical University, 50-367 Wroclaw, Poland; jacek.smereka@umed.wroc.pl; 2Institute of Outcomes Research, Maria Sklodowska-Curie Medical Academy, 00-001 Warsaw, Poland; lukasz.szarpak@uczelniamedyczna.com.pl; 3Henry JN Taub Department of Emergency Medicine, Baylor College of Medicine, Houston, TX 77030, USA; 4Department of Obstetrics and Gynecology, Unaizah College of Medicine and Medical Sciences, Qassim University, Unaizah 56219, Saudi Arabia; ishagadam@hotmail.com

**Keywords:** COVID-19 pandemic, pregnancy, homebirth

## Abstract

*Background and Objectives:* COVID-19 is a pandemic disease, and its unpredictable outcome makes it particularly dangerous, especially for pregnant women. One of the decisions they have to make is where they will give birth. This study aimed to determine the factors influencing the choice of place of delivery and the impact of the COVID 19 pandemic on these factors. *Materials and Methods*: The study was conducted on 517 respondents from Poland. The research methods comprised the authors’ own survey questionnaire distributed via the Internet from 8 to 23 June 2021. The survey was fully anonymous, voluntary, and addressed to women who gave birth during the pandemic or will give birth shortly. *Results:* A total of 440 (85.1%) respondents were afraid of SARS-CoV-2 infection. The most frequently indicated factors were fear of complications in the newborn, fear of intrauterine fetal death, and congenital disabilities in a newborn. A total of 74 (14.3%) women considered home delivery. The main factors that discouraged the choice of home birth were the lack of professional medical care 73.1% (*N* = 378), the lack of anesthesia 23.6% (*N* = 122), and the presence of indications for caesarean section 23.4% (*N* = 121). The possibility of mother–child isolation caused the greatest fear about hospital delivery. During the COVID-19 pandemic, pregnant women concerned about SARS-CoV-2 infection were more likely to consider home delivery than those without such fears. The most important factors affecting the choice of the place of delivery included the possibility of a partner’s presence, excellent sanitary conditions and optimal distance from the hospital, and the availability of epidural analgesia for delivery. *Conclusions:* Our study identifies the determinants of place of delivery during the COVID-19 pandemic. The data we obtained can result in the healthcare system considering patients’ needs in case of similar crisis in the future.

## 1. Introduction

The first coronavirus disease 2019 (COVID-19) cases were documented in December 2019 in Wuhan, China. This occurrence changed the world we used to know and started the most significant global health disaster of the 21st century [1,2]. As of 20 May 2022, the total number of confirmed cases of COVID-19 infection was 526,455,518 worldwide, with 6,286,610 deaths. Current countries with the highest number of confirmed cases are the United States and India, 83,141,628 and 43,131,822, respectively. In Poland, the numbers are 6,005,101 cases with 116,255 deaths and still are counting [3]. There are a few studies assessing the impact of COVID 19 on the course of pregnancy, especially in the context of the third trimester [4,5]. A total of 767 home deliveries took place in 2017 in Poland [6]. It was certainly influenced by the fact that, according to Kopacz et al., only 10% of Polish hospitals offer women the freedom to choose a position during childbirth, and only 27% of them allow free contact with the newborn after childbirth. Moreover, medical interventions such as episiotomy or amniotomy are often performed without medical indications and consent from the mother [7]. The result of the global pandemic could be observed in all areas of medical care. We noticed an increased incidence of out-of-hospital cardiac arrest, lower rates of successful resuscitation, and increased mortality. It has significantly impacted patient outcomes through decreased access to care and the reshaping of emergency medical response and hospital-based healthcare systems policies. Furthermore, attitudes toward resuscitation have also changed negatively, and providers were challenged with novel ethical dilemmas [8]. To the authors’ knowledge, there is currently no research assessing the impact of pandemics on the choice of the place of delivery, which gives pregnancy during the pandemic a new perspective. Many countries worldwide have responded to the COVID-19 pandemic by restricting the movement of citizens, obligating citizens to wear face masks, and transforming many hospitals into dedicated units prepared to treat COVID-19 patients only, which has affected the overall picture of health service availability and general public health [9,10]. These limitations also influenced pregnant patients who as a group have been susceptible to infections caused by other coronaviruses, including Severe Acute Respiratory Syndrome (SARS) and Middle East Respiratory Syndrome (MERS). In response, the International Federation of Gynecology and Obstetrics (FIGO) recommended routine pregnancy follow-up visits be suspended. As long as possible, it is advised to implement online consultations [11,12]. It is already clear that the pandemic has a significant impact on the human psyche, everyday living, and the quality of life [13,14,15]. The authors null hypothesis is that the determinants of place of delivery during the COVID-19 pandemic were multifactorial. The study aimed to extract the main determinants for choosing the place of delivery in the SARS-CoV2 pandemic.

## 2. Materials and Methods

The presented study is based on the authors’ own questionnaire. The survey was shared via social networking sites between 8 and 23 June 2021. This period occurred just after Poland’s so-called third wave of the coronavirus pandemic. Some gynecological hospitals were transformed into hospitals dedicated to patients who were SARS-CoV-2 positive only. The questionnaire consisted of 22 questions. Participation in the study was fully anonymous and voluntary. The study was addressed to women who gave birth during the pandemic or are about to give birth. The research group was recruited via Facebook groups and linked to the questionnaire shared on Instagram Stories. Participation in the study required informed consent. The right to leave the study was sustained at all times. The elaboration of the results was based on the statistical analysis of measurable (quantitative) and non-measurable (qualitative) features. The analysis of the relationships between the qualitative variables was carried out with the use of cross tables with the use of Chi2 tests, Likelihood ratio Chi2, and the exact Fisher test. The strength of the compounds was measured using the Phi Yule coefficient. A correlation between quantitative variables was verified using Spearman’s rho test. A significance level of *p* < 0.05 was adopted, indicating the presence of statistically significant relationships or differences. Statistical analysis was performed using the SPSS 26 software. The research was approved by the Institutional Review Board of the Polish Society of Disaster Medicina (Approval no. 02.07.2021.IRB).

## 3. Results

The study involved 517 respondents residing in Poland. The average age of the respondents was 30 years, the youngest was 18 years old, and the oldest was 42 years old.

A detailed description of the research group is presented in Table 1.

One of the most important statistics to obtain in our study was the answer to the question about the factors influencing the choice of delivery place. As a result of the analysis, it was observed that the most frequently chosen answers to this question were: the possibility of a family birth (56.3%; 291 people), very good sanitary conditions (39.5%; 204 people), optimal distance from the hospital (39.3%; 203 people), and the opinion of other patients (36.4%; 188 people) (Table 2).

During the study, the respondents were also asked about fears of getting SARS-CoV-2 during pregnancy and if the SARS-CoV-2 pandemic has/had an impact on the choice of delivery place.

Some 85% of the respondents admitted that they were afraid of being infected with SARS-CoV-2, and only 15% were not afraid.

As a result of the analysis, it was observed that 62.7% (324 women) of the group answered negatively to the question asking if the pandemic had any impact on the choice of delivery place; however, the remaining 37.7% (193 people) admitted it did (see Figure 1 and Figure 2).

Apprehension associated with COVID-19 was multifactorial and is presented in Figure 3.

The impact of the pandemic on pregnancy check-ups was also analyzed. The hardest part for pregnant women was the absence of their partner during the visits (Figure 4).

Some 14.3% of patients (*N* = 74) have considered a home delivery. Among the patients who did not consider home birth, the main factors were the lack of specialist medical care 73.1% (*N* = 378), no possibility of anesthesia 23.6% (*N* = 122), the presence of indications for caesarean section 23.4% (*N* = 121), and the lack of local conditions to having home childbirth 15.7% (*N* = 81). In turn, for patients considering home delivery, the decisive factors are presented in Table 3.

The relationship between the fear of SARS-CoV-2 infection during pregnancy and the consideration of having a home birth was also analyzed (Table 4). As a result of the analysis, it was possible to confirm the significant statistical relationship.

It was observed that among people who were afraid of infection, the percentage of people who considered giving birth at home (11.82%; 52 people) was significantly lower than those who were not afraid of infection (28.57%; 22 people). The relationship was substantiated by the test: χ2 = 15.00; df = 1; *p* < 0.001; ϕ = −0.170; *p* < 0.001.

Additionally, we assessed the relationship between age and the number of delivered vaginal births combined with the consideration of having a home birth (Table 5).

The conducted analysis did not show any statistically significant connection between considering home birth and the age of the respondents (U = 14,489.5; *p* = 0.103), but there was a statistically significant relationship between the number of delivered births and considering a home birth.

Another result is the relationship analysis between considering home birth and cesarean delivery (Table 6). The analysis shows that women who did not have a caesarean section more frequently considered giving birth at home 61 (17.2%) than those who chose this way of delivery 13 (8%). Analysis by Fisher’s exact test showed that this relationship is statistically significant.

The elaboration of the results was based on the statistical analysis of measurable (quantitative) and non-measurable (qualitative) features. The analysis of the relationships between the qualitative variables was carried out with cross tables with the use of Chi2 tests, Likelihood ratio Chi2, and the exact Fisher test. The strength of the compounds was measured using the Phi Yule coefficient. A correlation between quantitative variables was verified using Spearman’s rho test. A significance level of *p* < 0.05 was adopted, indicating the presence of statistically significant relationships or differences. Statistical analysis was performed using the SPSS 26 software.

## 4. Discussion

The authors are not aware of the existence of similar work. However, many studies have explored determinants and factors that influence the choice of delivery place, especially the choice between in-hospital and out-of-hospital births; none of them looked into those determinants during the pandemic. For women choosing childbirth at a hospital, the most essential thing was perceptions of safety, choice of medicalization and the option for pain relief, or the availability of medical care [16,17]. Women choosing non-hospital births emphasized a desire for individual care of the midwife, a familiar environment, control over the birth process, and more involvement from partners, children, and family [18]. We know that women have to extract information from various sources to form their views. They were using the Internet, friends’ recommendations and experiences, antenatal and birth preparation classes, and their own experiences of delivery [19]. Perinatal and neonatal mortality rates are different in many studies. Some studies show higher among planned home deliveries than among hospital deliveries [20,21,22]. In contrast, there are also reports showing no significant difference between them [23,24,25,26]. The number of women who deliver in hospitals grew with advanced medical care and a higher ability to rescue newborns and mothers [27]. It allows for providing the necessary care in the case of perinatal complications. The main reason for this reduction in maternal and perinatal mortality is blood transfusions, antibiotics, and safe anesthesia. [28,29]. The reduction in maternal mortality has made many people believe that pregnancy and delivery are now safe. This belief has led to demand for a return to home birth from many social groups and is treated as a woman’s right [30]. Home birth supporters are driven by three main factors: the right to choose, high hospitalization cost, and the possibility of having a free birth, which may be dangerous. On the other hand, opponents argue that physiological childbirth can always turn pathological, and transportation to the hospital in time might not be possible [31]. The impact of COVID-19 infection can be divided into infection in the early (up to 12 weeks of pregnancy) and late stages (after 24 weeks of pregnancy). Seasonal influenza has been associated with a higher rate of spontaneous miscarriage [32]. A similar relationship is sought in COVID-19 infection, but there is still no hard evidence. Cosma, in his study, analyzed the cases of 225 early pregnancy patients with confirmed COVID-19 infection [33]. The research showed no connections between severe acute respiratory syndrome in coronavirus infection during the first trimester and early pregnancy loss. The situation is entirely different in late pregnancy. The largest cohort study from the United States included 91,412 women, of which 8207 were pregnant [34]. Pregnancy was associated with an increased risk of hospitalization among COVID-positive patients (RR, 5.4; 95% CI, 5.1– 5.6) and a higher need for mechanical ventilation (RR, 1.7; 95% CI, 1.2–2.4). However, there was no significant peak in mortality (RR, 0.9; 95% CI, 0.5–1.5). The systematic review and meta-analysis by Capobianco et al. [35] reported preterm births, neonatal pneumonia, and respiratory distress syndrome in infants born of COVID-19-positive mothers. Furthermore, some studies report other complications such as premature rupture of membranes (PROM), preterm deliveries, lymphopenia, pre-eclampsia, placenta previa, hypothyroidism, oligohydramnios polyhydramnios, fetal distress, increased cesarean deliveries, abnormal umbilical cord, and sinus tachycardia [35,36,37,38,39,40,41,42].

The study of the factors that influence the choice of the place of delivery seems to be important for both medical staff and patients. Medical staff taking care of a pregnant patient should have information about what is most important for the woman giving birth. During unusual times such as war, pandemic, or economic crisis, it should still be important to be able to provide the best care for the pregnant woman. Factors important for pregnant women for choosing delivery place during the COVID-19 pandemic were identified in our study. However, we do not know what we will have to face in the future, which can be seen in the example of women giving birth in Ukraine in connection with the war in their country [43,44].

This study is innovative and describes the relationship between the pandemic and the pregnant patients’ choice of perinatal care. Nonetheless, our research is not without limitations. The most fundamental issue is that our data collection was conducted via the Internet. Nevertheless, that seems to be the only safe data collection tool in the pandemic. Moreover, studies confirm that people feel more comfortable during online surveys [45]. Another limitation is the relatively small research group and the fact that it targets the selected population—Polish pregnant women.

The null hypothesis of the multifactorial effect on the choice of place of delivery during COVID-19 pandemic was confirmed in the study. Some of these factors were more important, some less.

Further research should focus on the analysis of factors influencing the choice of the place of delivery depending on the various types of risks that may await us in the future.

## 5. Conclusions

The SARS-CoV-2 pandemic has changed people’s perception of the world. This special time also affects pregnant women. Patients with concerns about SARS-CoV-2 infection were more likely to consider home delivery than those without such fears. Our study highlights what matters for pregnant women when choosing a place for delivery: the presence of a partner and good enough sanitary conditions. They missed the most during their pregnancy because of the possibility of their partner’s presence during the pregnancy check-ups. The whole care system should be revised to be prepared for possible future problems and to meet patients’ needs (including pregnant women).

## Figures and Tables

**Figure 1 medicina-58-00831-f001:**
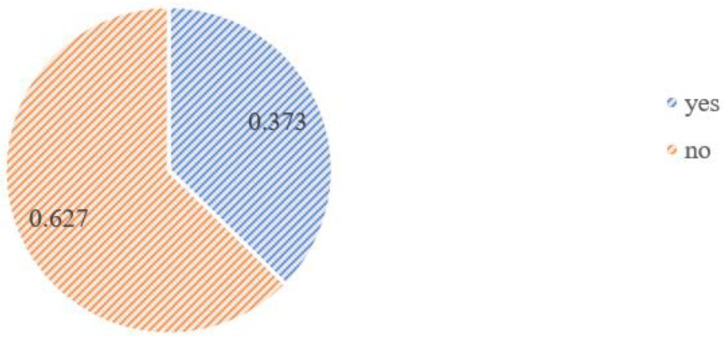
The frequency analysis graph for the answer to the question “Does the SARS-CoV-2 pandemic have/had an impact on the choice of delivery place?”.

**Figure 2 medicina-58-00831-f002:**
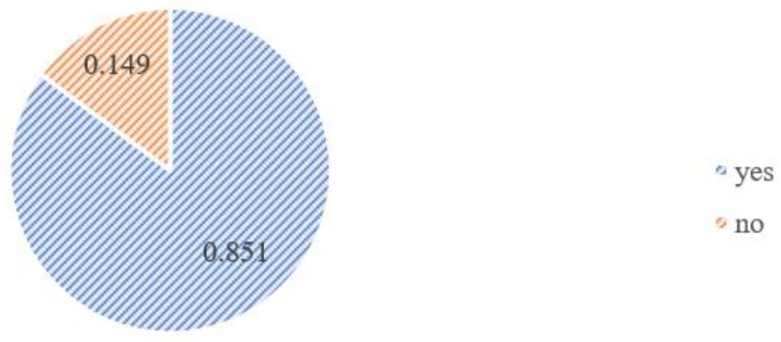
The frequency analysis graph for the answer to the question “Were you afraid of SARS-CoV-2 infection during your pregnancy?”.

**Figure 3 medicina-58-00831-f003:**
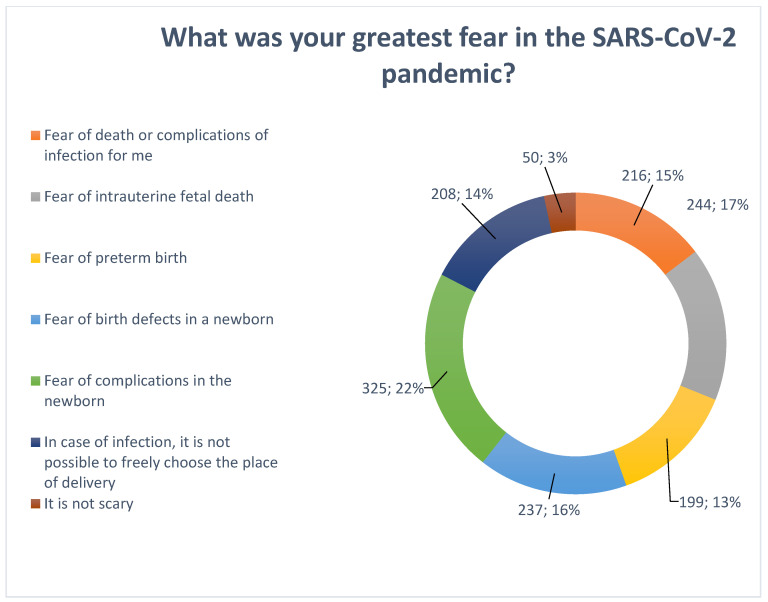
Factors that cause anxiety in pregnant women about the risk of COVID-19 infection.

**Figure 4 medicina-58-00831-f004:**
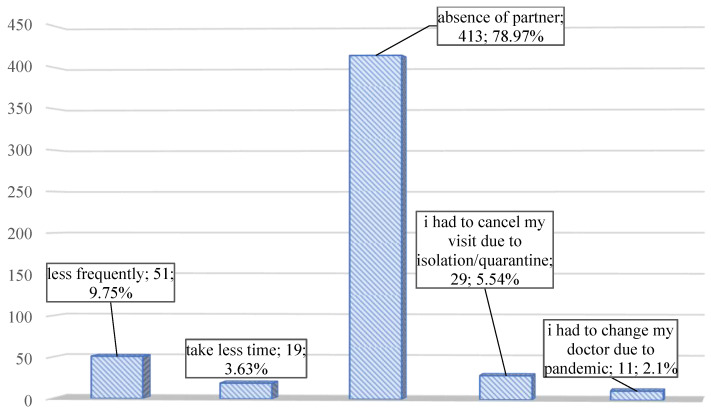
The frequency analysis graph for the answer to the question “How has the SARS-CoV-2 pandemic affected your pregnancy visits?”.

**Table 1 medicina-58-00831-t001:** Characteristics of the research group.

	Frequency	Proportions (%)
Place of residence		
Countryside	131	25.3
City of up to 50,000 residents	73	14.1
City of up 50,000–25,0000 residents	80	15.5
City of above 250,000 residents	233	45.1
Marital status		
Single	74	14.3
Partnership	12	2.3
Married	426	82.4
Divorced	5	1.0
Education status		
Higher education	430	83.2
Secondary education	80	15.5
Vocational education	5	1.0
Primary education	2	0.4

**Table 2 medicina-58-00831-t002:** The results of the frequency analysis for the answer to the question “Choose the 3 most important factors that affect your choice of the place of delivery”.

	*N*	%
Family childbirth possible	291	56.3
Very good sanitary conditions	204	39.5
Optimal distance from the hospital	203	39.3
Opinion of other patients	188	36.4
Free choice of birthing position	130	25.1
Possibility of epidural anesthesia	130	25.1
Childbirth according to nature	83	16.1
Possibility of choosing a dedicated midwife	63	12.2
Waterbirth	20	3.9
A higher degree of referentiality in neonatal care	0	0.0

*N*—group size, %—a percentage of the group.

**Table 3 medicina-58-00831-t003:** The frequency analysis results in the answer to the question, “If you considered giving birth at home, what factors impact your decision?”.

	*N*	%
Fear of infection with SARS-CoV-2 in the hospital	0	0.0
The possibility of giving birth without, in my opinion, unnecessary medications	28	5.4
Distrust of hospital staff	18	3.5
Fear of isolating the mother from the child after childbirth	50	9.7
Intimate conditions	48	9.3

**Table 4 medicina-58-00831-t004:** The results of the analysis of the relationship between the fear of SARS-CoV-2 infection during pregnancy and the consideration of having a home birth.

Were You Worried about Getting SARS-CoV-2 during Pregnancy?		Have You Considered Giving Birth at Home:	
		No	Yes
No	*N*	55	22
	%	71.43%	28.57%
Yes	*N*	388	52
	%	88.18%	11.82%

**Table 5 medicina-58-00831-t005:** The results of the analysis of the relationship between age and number of delivered vaginal births combined with the consideration of having a home birth.

Have You Considered Giving Birth at Home:		*N*	M	SD	U	*p*
Age of patient	No	444	30.11	3.62	14,489.5	0.103
	Yes	74	29.47	3.95		
Natural births in past	No	444	0.57	0.75	10,885	0.001
	Yes	74	1.12	0.95		

**Table 6 medicina-58-00831-t006:** The results of the analysis of the relationship between considering home birth and cesarean delivery.

Did Your Pregnancy End with Cesarean Section?		Have You Considered Giving Birth at Home?	
		No	Yes
No	*N*	294	61
	%	82.8%	17.2%
Yes	*N*	150	13
	%	92%	8%

Fisher’s exact test = 7.734; df = 1; *p* = 0.005; Cramer V = 0.122; *p* = 0.005.

## Data Availability

Data can be obtained individually after contacting mateusz.strozik@student.umw.edu.pl.

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
