# Peer review of "Determinants of Place of Delivery during the COVID-19 Pandemic—Internet Survey in Polish Pregnant Women"

_medicina, 2022, doi:10.3390/medicina58060831_

Round 1
Reviewer 1 Report
Find below the summary of my observation of the article.
The authors studied the determinants of place of delivery during the COVID-19 pandemic among Polish pregnant women using an internet survey.
Seventy-four (14.3%) of the women considered home delivery. The women who are concerned about SARS-Cov-2 infection were more likely to consider home delivery than those without such fears.
A limitation of the study is the fact that it involves only Polish women and it is difficult to generalize to other populations. With the use of an internet survey, it would have been possible to extend the questionnaires to other parts of the world such that the results become more generalizable. Internet is present almost in all parts of the world and the pandemic affected the whole globe.
It is important to state if ethical approval for the study was obtained and if not why not.
Table 5 needs to be clear. The question 'what is your age is not reflected on the table.
Author Response
Dear Editor,
First of all we would like to thank you very much for taking time to review our article.
Before starting the study, the approval of the bioethical committee was obtained by Institutional Review Board of the Polish Society of Disaster Medicina (Approval no. 02.07.2021.IRB). Information was already included in the article.
We changed the look and information in table 5 so we hope it is more clearly now.
Thanks again for your time
Reviewer 2 Report
The study is very interesting and has a scientific value, however the authors should address the following points:
* The abstract is well-written, however word count should be checked per the journal's guidelines.
* The authors should use passive voice in manuscript preparation and editing.
* The authors should clarify the study objectives and add null hypothesis at the end of the introduction section.
* What was the "original questionnaire" that was referred to by authors?
* Was the questionnaire validated? pilot study done?
* The authors should mentioned the IRB approval code and along with the approving institution.
* The authors should add directions for future research.
* The authors should state whether their hypothesis is accepted or rejected.
* The authors should add a section reflecting the clinical significance of their findings.
* The abstract should be summarized in bullets.
Author Response
Dear Editor,
First of all we would like to thank you very much for taking time to review our article.
In relation to your comments:
* The abstract is well-written, however word count should be checked per the journal's guidelines.
We corrected the abstract to meet the criteria of the journal. The word count is 291 (the maximum is 300)
* The authors should use passive voice in manuscript preparation and editing.
Thank you for your attention, we have changed to passive form in places that are visible to us
* The authors should clarify the study objectives and add null hypothesis at the end of the introduction section.
According to your suggestion, we added the null hypothesis at the end of the introduction and also we edited the research goals.
* What was the "original questionnaire" that was referred to by authors?
When we use the term original, we meant the authors' own questionnaire. We removed original and replaced it with authors' own questionnaire.
* Was the questionnaire validated? pilot study done?
Our survey is a proprietary survey containing mainly social, demographic and clinical data. The questionnaire did not have the characteristics of a psychological questionnaire, therefore the terms and concepts used were unambiguous. In our opinion, it did not require validation. Due to the fact that it was a short study that covered a specific exposure period, conducting a pilot study could lead to the extension of the entire study process. It was important for us to obtain information during the pandemic, when the fear of disease was greatest.
* The authors should mentioned the IRB approval code and along with the approving institution.
Before starting the study, the approval of the bioethical committee was obtained by Institutional Review Board of the Polish Society of Disaster Medicina (Approval no. 02.07.2021.IRB). Information was already included in the article.
* The authors should add directions for future research.
Thank you for your attention, in discussion we added information about the direction of the next research "Further research should focus on the analysis of factors influencing the choice of the place of delivery, depending on the various types of risks that may await us in the future."
* The authors should state whether their hypothesis is accepted or rejected.
We also referred to the null hypothesis and confirmed its assumptions
* The authors should add a section reflecting the clinical significance of their findings.
Following your suggestion, we've added a paragraph to the discussion on the clinical significance of our findings
* The abstract should be summarized in bullets.
We corrected the abstract to meet the criteria of the journal.
Thank you again for your suggestions and advices